# Regulation of Amylose Content by Single Mutations at an Active Site in the Wx-B1 Gene in a Tetraploid Wheat Mutant

**DOI:** 10.3390/ijms23158432

**Published:** 2022-07-29

**Authors:** Yulong Li, Hassan Karim, Bang Wang, Carlos Guzmán, Wendy Harwood, Qiang Xu, Yazhou Zhang, Huaping Tang, Yunfeng Jiang, Pengfei Qi, Mei Deng, Jian Ma, Jingyu Lan, Jirui Wang, Guoyue Chen, Xiujin Lan, Yuming Wei, Youliang Zheng, Qiantao Jiang

**Affiliations:** 1State Key Laboratory of Crop Gene Exploration and Utilization in Southwest China, Sichuan Agricultural University, Chengdu 611130, China; 2020212029@stu.sicau.edu.cn (Y.L.); hassankarim@stu.sicau.edu.cn (H.K.); wangbang@stu.sicau.edu.cn (B.W.); xuqiang1264700418@163.com (Q.X.); yazhou14716@sicau.edu.cn (Y.Z.); tanghuapin2019@126.com (H.T.); jiangyunfeng@sicau.edu.cn (Y.J.); pengfeiqi@hotmail.com (P.Q.); dengmei105@163.com (M.D.); jianma@sicau.edu.cn (J.M.); jingyulan@sicau.edu.cn (J.L.); jirui.wang@gmail.com (J.W.); guoyuech74@hotmail.com (G.C.); lanxiujin@163.com (X.L.); ymwei@sicau.edu.cn (Y.W.); ylzheng@sicau.edu.cn (Y.Z.); 2Triticeae Research Institute, Sichuan Agricultural University, Chengdu 611130, China; 3Departamento de Genética, Escuela Técnica Superior de Ingeniería Agronómica y de Montes, Edificio Gregor Mendel, Campus de Rabanales, Universidad de Córdoba, 14071 Cordoba, Spain; carlos.guzman@uco.es; 4John Innes Centre, Norwich Research Park, Norwich NR4 7UH, UK; wendy.harwood@jic.ac.uk

**Keywords:** tetraploid wheat, waxy gene, protein structure, ADPG binding pocket, starch-binding capacity, amylose content

## Abstract

The granule-bound starch synthase I (GBSSI) encoded by the waxy gene is responsible for amylose synthesis in the endosperm of wheat grains. In the present study, a novel *Wx-B1* null mutant line, M3-415, was identified from an ethyl methanesulfonate-mutagenized population of Chinese tetraploid wheat landrace Jianyangailanmai (LM47). The gene sequence indicated that the mutated *Wx-B1* encoded a complete protein; this protein was incompatible with the protein profile obtained using sodium dodecyl sulfate–polyacrylamide gel electrophoresis, which showed the lack of Wx-B1 protein in the mutant line. The prediction of the protein structure showed an amino acid substitution (G470D) at the edge of the ADPG binding pocket, which might affect the binding of Wx-B1 to starch granules. Site-directed mutagenesis was further performed to artificially change the amino acid at the sequence position 469 from alanine (A) to threonine (T) (A469T) downstream of the mutated site in M3-415. Our results indicated that a single amino acid mutation in Wx-B1 reduces its activity by impairing its starch-binding capacity. The present study is the first to report the novel mechanism underlying Wx-1 deletion in wheat; moreover, it provided new insights into the inactivation of the waxy gene and revealed that fine regulation of wheat amylose content is possible by modifying the GBSSI activity.

## 1. Introduction

Wheat is a crop of major world significance [1]. In particular, tetraploid durum wheat (*Triticum turgidum* L. ssp. durum (Desf.) Husn, 2n = 4x = 28, AABB) is an important cereal crop with an annual worldwide production of over 40 million tons [2]. Tetraploid durum wheat provides the raw material for semolina, pasta, couscous, and several other dishes of the Mediterranean tradition [3], and it is the second most important wheat species after common wheat [4]. Starch is the major component of wheat grain at 70% of the dry weight [5], consists of two classes of glucose polymers, amylose and amylopectin, which are defined based on their α-1, 4 and α-1, 6 linkages and molecular weight. Amylose features a low degree of polymerization (DP < 5000) as compared with amylopectin (DP > 5000) [6,7]. The amylose to amylopectin ratio determines some of the functional and physicochemical properties of starch, thereby affecting the quality of durum wheat [8,9].

Starch synthesis is a complex process involving multiple enzymes and isoforms. The synthesis of amylopectin requires three important classes of enzymes: starch branching enzymes (SBE), starch debranching enzymes (DBE), and starch synthase (SS) [6]. Amylose synthesis requires only granule-bound starch synthase I (GBSSI or waxy protein) enzyme [10,11]. Durum wheat has two waxy proteins encoded by Wx-A1 and Wx-B1 [12]. High-amylose wheat varieties with a lower glycemic index that is beneficial for human health have been produced over the last decades by modifying the SBE, DBE, and SS genes. Unfortunately, the end-use quality of these wheat varieties has generally been reported to be disappointing [13]. Conversely, wheat varieties with lower amylose content have been reported to enhance the end-use quality of specific wheat products, such as noodles, bread, and pasta [14,15]. In a previous study, null alleles of the Wx-1 genes were used for the production of low amylose wheat [16].

GBSSI, soluble starch synthase, and glycogen synthase (GS) belong to the GT5 (GT-B) glucosyltransferase family. These proteins share a common structure existing in the C-terminal and N-terminal domains containing donor- and acceptor-binding sites, respectively [17,18,19]. These two domains exhibit a crucial role in the function of enzymes as active sites are included within them and nonsynonymous substitutions in these active sites can affect protein function [20]. The first oligosaccharide-bound GS structure facilitated the analysis of the contributions of the corresponding amino acid residues in GBSSI to its activity [21]. Based on the structural and functional similarities among the GT-B family enzymes, formulating hypotheses on the locations of the key amino acid residues that regulate amylose content in waxy proteins was possible [22].

Over the last few decades, chemicals have been widely used for mutagenesis in crops. In particular, ethyl methanesulfonate (EMS) has been reported to cause mutations in C–T and G–A substitutions with a high frequency [23,24,25]. In our previous work, we generated an EMS-mutagenized tetraploid wheat population [26], and in the present study, we identified a Wx-B1 null mutant from this population. The sequence of the Wx-B1 gene of the mutant encoded a complete protein, which was incompatible with the protein profile obtained via sodium dodecyl-sulfate polyacrylamide gel electrophoresis (SDS-PAGE) showing the lack of Wx-B1 protein in the mutant line. This implies that the absence of Wx-B1 protein may be due to a novel mechanism. This study describes the characterization of the novel molecular mechanism underlying *Wx-B1* deletion in wheat and the effect of the mutation in *Wx-B1* on starch-binding capacity, GBSSI activity, and amylose content.

## 2. Results

### 2.1. Identification of the Wx-B1 Null Mutant

An M2-mutagenized population of cultivar LM47 was screened for variability in waxy proteins using SDS-PAGE. Three grains were chosen from each plant from the mutagenized population, and M2-415 was identified as a mutant lacking Wx-B1 protein. The remaining M2-415 grains were grown in a greenhouse to produce the next generation, the lack of Wx-B1 was confirmed in the M3 generation of the M3-415 mutant line (Figure 1a).

### 2.2. Identification of Causal Mutations in M3-415

Two pairs of PCR primers were designed to amplify the open reading frame (ORF) of *Wx-B1* from both the wild-type (WT) and mutant lines and investigate the molecular mechanism underlying Wx-B1 protein deletion in the M3-415 mutant. The *Wx-B1* gene sequence of mutant M3-415 was aligned to the gene sequence of the WT line. The results showed only one single-nucleotide mutation in the M3-415 *Wx-B1* gene consisting of a G to A change at 2088 bp (named G2088A) downstream of the ATG start codon in exon 8. This mutation changed the codon from GGC to GAC and led to a residue change from glycine to aspartic acid in the deduced protein sequence; however, a stop codon was not observed (Appendix A). The gene sequence of *Wx-B1* was incompatible with the protein profile obtained via SDS-PAGE that showed the lack of Wx-B1 protein in the mutant line.

### 2.3. Measurement of Total Grain Proteins

The M3-415 gene sequence indicated that it could encode a complete Wx-B1 protein, although it was inconsistent with the protein composition of mature grains. Therefore, developing grains of M3-415 were selected to identify Wx-B1 protein using SDS-PAGE (Figure 1b). As observed in mature grains, locating Wx-B1 protein in developing grains was also not possible. In addition, the total protein level between mutant and parental lines was measured (Figure 1c), and the presence of Wx-B1 and remaining proteins was revealed. Thus, we assumed that Wx-B1 might be present in the grains, as evidenced by the results of total grain protein, but without being bound to starch granules. As a result, detecting Wx-B1 protein in M3-415 via SDS-PAGE was probably not possible. Hence, it was speculated that this mutation might have affected the protein structure in the mutant line, which in turn might have caused the lack of binding to starch granules.

### 2.4. Modification of the ADPG Binding Pocket in the M3-415 Mutant

We hypothesized that the mutated Wx-B1 protein may have lost the ability to bind to starch granules. Hence, the protein tertiary structure between the WT and mutant line was compared using the Swiss model. The three-dimensional protein structure showed that single amino acid substitutions in M3-415 were located at the edge of the ADPG binding pocket (Figure 2a,b). The Plant Protein Variation Effect Detector (http://www.ppved.org.cn, accessed on 1 January 2021) was used to detect the effect of mutational sites in M3-415 and predict the changes in functionality caused by the single amino acid substitutions. Among the single amino acid substitutions found in Wx-B1 protein, G470D was predicted to be functional (Table 1). In addition, the impacts of the amino acid substitutions on the GBSSI structure were examined through the Swiss-pdb viewer, which showed that Asp contained a carboxy I group. Most significantly, an enhanced van der Waals force was produced at D470, hindering the entry of ADP into the binding pocket (Figure 2c,d). Therefore, we speculated that the likely effect of the mutation at this site was to prevent GBSSI from binding to starch granules.

### 2.5. Evaluation of Starch-Binding Capabilities

The ADPG binding pocket plays an important role in binding GBSSI to starch granules and contributes significantly to the GBSSI activity. The binding capability of GBSSI was analyzed in WT and mutant lines (Figure 3a,b), and the results showed that the binding ability of M3-415 was significantly reduced in the mutant line compared with the WT line, which was denoted by the presence of a strong band in the supernatant showing that the mutant line could not bind to starch granules. In addition, to verify the effect of this domain on starch-binding capabilities, site-directed mutagenesis of the amino acid was performed near the ADPG binding pocket functional domain in the WT line. Alanine (A) was substituted with threonine (T) at the amino acid sequence position 469 (A469T) downstream of the mutated site in M3-415. Indeed, the results of the site-directed mutagenesis confirmed that the A469T mutation near the ADPG binding pocket destroyed the starch-binding capability of GBSSI (Figure 3b). The results were found to be consistent for A469T and M3-415, as confirmed by further starch-binding experiments. As previously hypothesized, it was confirmed that the starch-binding capabilities of the M3-415 mutant decreased.

### 2.6. Expression Pattern of the Wx Gene

The expression patterns of the *Wx-B1* gene, preferentially expressed between the M3-415 mutant and WT lines at different developmental stages (5, 10, 15, 20, and 25 d after fertilization [DAF]), were compared in a developing endosperm using qRT-PCR. The analysis revealed no significant differences in the expression patterns in either of the two lines (Figure 3c). Hence, the single-base substitution in M3-415 did not affect the expression level of *Wx-B1*.

### 2.7. Measurement of Enzymatic Activity in the M3-415 Mutant

GBSSI enzyme activity assays were performed in the immature endosperm of the M3-415 mutant and WT lines to quantify the effect of M3-415 mutations on GBSSI activity. The enzymatic activity was measured in developing grains. Three replicates for each line were collected at 10 days post anthesis. A 60% reduction in the GBSSI activity level was observed in the M3-415 mutant compared with that in the WT line (Figure 3d).

### 2.8. Physicochemical Properties of Starch in M3-415 Grains

The physicochemical properties of starch in the M3-415 mutant line was determined and are presented in Table 2. This line showed lower percentages of total starch (5.87%) (Figure 4a) and amylose content (3.01%) (Figure 4b) than the WT line. The thermal properties determined using a differential scanning calorimeter (DSC) assay, and the physicochemical parameters including To, Tp, Tc, and ΔH. There was no significant in the To between the two lines. However, the Tp, Tc and ΔH were significantly increased in the mutant line compared with the WT line (Table 2). The results of urea digestion are presented in Figure 4c,d. Both the mutant and WT lines were solubilized in 0–6 M urea, and no differences were observed between them.

### 2.9. Starch Granule Characterization

Scanning electron microscopy (SEM) was performed to determine whether the Wx-B1 mutation affected the morphology of starch granules, and the results are presented in Figure 5. Compared with the starch particle morphology in the WT line, that in the M3-415 line showed rough surfaces, and the type A granules showed a higher degree of incompleteness and edge damage than that in the WT line. In addition, the percentage of type A and type B starch granules in the M3-415 and WT lines were measured; no significant differences were observed between the two lines in terms of number percentages. However, the volumes of the A- and B- type starch granules in the M3-415 line were larger than those in the WT line (Table 3).

## 3. Discussion

EMS mutagenesis has been widely used for the improvement of numerous crops, such as rice, barley, wheat, and soybean [27]. Using this technique, the *Wx-1* gene has been successfully modified in hexaploid and tetraploid durum wheat varieties, resulting in mutants generally lacking one Wx protein [28]. Some natural null mutants were identified from tetraploid and hexaploid wheat varieties and landraces of different geographic origins [29]. In *Wx-B1b*, the most prevalent allele of the known *Wx-B1* null alleles, silencing of the *Wx-B1* gene is caused by the deletion of an approximately 67-kb fragment that includes the entire *Wx-B1* gene coding region [30]. Other *Wx-B1* alleles, such as *Wx-B1k* and *Wx-B1m*, were found in *T. compactum* and Indian dwarf wheat. As sequence alignment revealed, the deletion of the Wx-B1 protein subunit was the result of an insertion of four bases (GCTA) in the seventh exon in *Wx-B1k* and a deletion of four bases (AACA) in the second exon in *Wx-B1m* [31]. We have previously characterized a novel *Wx-B1* allele in which a 2178-bp transposon fragment was inserted within the tenth exon, leading to the absence of Wx-B1 protein, and we revealed a different molecular mechanism of the *waxy* gene [32]. In addition, we also identified a Wx-A1 null mutant from an EMS-mutagenized population of common wheat cv. In SM126, a single-nucleotide mutation substituting A for G at the splicing site led to incorrect RNA splicing and ultimately resulted in the loss of Wx-A1 protein [33]. In the present study, we describe a novel molecular mechanism, hitherto unreported in wheat, that leads to the loss of Wx-B1 in an EMS mutant tetraploid wheat line.

The examination of the local configuration of Gly470 in a tertiary model based on the published structure of rice GBSSI, a very close homolog of durum wheat GBSSI [34], did not provide an immediate explanation as to why GBSSI in M3-415 should exhibit low activity. GBSSIa belongs to the GT5 family of glycosyltransferases in the CAZy database [34] that share the characteristic GT-B (double Rossmann) fold. In GBSSIa, the interface between two Rossmann folds forms the catalytic site, which allowed the identification of an ADP molecule in the rice GBSSI structure [18]. Gly470 is neither positioned in this catalytic site nor involved in any putative substrate binding; instead, it is located at the edge of the ADPG binding pocket. Our results indicated that single amino acid substitutions in the functional domain affected the functions of Wx-B1 by reducing the ability of GBSSI to bind to starch granules. This finding is consistent with the general consensus that GBSS-type enzymes are tightly bound to starch granules within the internal granular matrix [35,36]. In this study, M3-415 exhibited the lowest starch-binding capability, explaining the marked loss of GBSSI activity. In summary, the single-base mutation adjacent to the ADPG binding pocket affected the ability of GBSSI to bind to starch granules and ultimately reduced GBSSI activity, which affected amylose production.

GBSSI is the predominant protein in starch granules [36]. While soluble starch synthases and SBE form complexes [37,38], GBSS-type enzymes do not contain any identified starch-binding modules, and they do not interact with the soluble starch synthases or SBE contained in these modules in the abovementioned complexes. In *Arabidopsis* leaves, amylose synthesis depends on a GBSS-type enzyme that targets transient starch granules via interaction with the protein targeting to starch (PTST), which contains a carbohydrate-binding module. PTST-mutant *Arabidopsis* plants synthesize wax-like transient starch in their leaves due to loss-of-function [39]. A PTST gene homolog exists in the tetraploid wheat genome. Therefore, the same mechanism may be necessary for the correct targeting of GBSSI to store starch granules and synthesize amylose in grain endosperm. Moreover, the findings in *Arabidopsis* plants demonstrate that mutations in genes other than *GBSS* can contribute to the waxy phenotype. Given that the binding of GBSS to PTST-type proteins is necessary for the correct targeting of GBSS and amylose synthesis [39], a possibility exists that the amino acid substitution in GBSSI in M3-415 prevents binding to a putative PTST protein, which in turn causes GBSSI to be incorrectly targeted. However, the binding of GBSS to PTST in *Arabidopsis* was mediated through a coiled-coil domain identified in the distal helix, which is not located in the structural area of the amino acid substitutions of M3-415.

The gelatinization properties of starch are closely related to its composition and morphology. The gelatinization of waxy starch requires more energy and higher temperatures than those required for the gelatinization of normal starch [40,41]. Gelatinization enthalpy (ΔH), a measure of the overall crystallinity of amylopectin, is negatively correlated with amylose content [42]. In addition to amylose content, the amylopectin branch chain-length distributions and starch granule size are other key factors affecting gelatinization [43]. Previous studies revealed that enthalpy of gelatinization (ΔH) is positively correlated with the the volumes of starch granules [44]. In this study, the M3-415 has lower amylose content, higher peak temperature and gelatinization enthalpy, this can be explained that the volumes of A-type starch granules and B-type starch granules in the mutant line were significantly increased than those in the WT line. In addition, the presence of pitting in the microstructure of the M3-415 starch granules, this has also been observed in waxy barley [45].

Durum wheat is the preferred raw material for pasta production, and cooking quality is one of the most important criteria in assessing the suitability of durum semolina for this purpose. The influence of the protein content and composition of durum wheat on the cooking quality of pasta has been extensively studied, but limited information is available on the effects of starch on the texture of cooked pasta [46,47]. Reconstitution experiments using starch from nondurum sources, e.g., waxy maize and wheat starch, showed a positive correlation between pasta quality and the amylose content of starch [48,49]. The loss of Wx protein directly affects amylose content. A previous study reported that low amylose content in wheat starch improved the texture, loaf volume, and shelf life of bread [50]. In addition, substitution of nonwaxy whole-wheat flour (WWF) with partial or full waxy WWF could modify the interaction among flour composition and influence the quality of noodle products [51]. Partial waxy durum wheat varieties appear to have an advantage over the normal types in terms of lower cooking loss, a widely held indicator of pasta cooking quality [15]. Therefore, the use of partial tetraploid waxy wheat could be recommended in pasta production. Here we proposed that the M3-415 mutant as partial waxy durum wheat might contribute to a better end-use quality of pasta.

## 4. Materials and Methods

### 4.1. Materials

In our previous study [26], an EMS-mutagenized population of Chinese durum wheat landrace Jianyangailanmai (LM47) was generated, and some waxy mutant lines were identified from this mutant population. In the present study, SDS-PAGE was used to detect more waxy mutants, and the M2-415 mutant with Wx-B1 protein deletion was identified. A total of 30 plants lacking Wx-B1 protein in the M3 generation (M3-415) were selected and grown in a greenhouse environment (light/dark: 16/8 h, day and night temperatures: 24/20 ℃, illuminance: 350 µ E·m^2^·s^−s^). Each plant exhibited at least three spikes, and all M4 plants were harvested separately from the M3 plants and stored in separate bags.

### 4.2. Electrophoresis of Wx Proteins and Extraction of Total Grain Proteins

Starch granules were prepared according to previously published methods [52]. Whole flour obtained from single mature wheat grains was mixed with 700 µL of protein extraction buffer I [53] Mm Tris–HCl, pH 6.8, 2.3% (*w*/*v*) SDS, 5% (*v*/*v*) β-mercaptothion, 10% glycerol], and the homogenate was centrifuged at 4 °C and 120,000× *g* for 5 min. Then, the samples were washed twice with 800 µL of double distilled water and acetone. Subsequently, the residue was mixed with 0.1 mg/µL of buffer I containing 0.005% (*w*/*v*) bromophenol blue, heated in a boiling bath for 5 min, and again centrifuged as mentioned above. The starch granule-bound proteins were analyzed via SDS-PAGE in a 12% acrylamide gel [31]. Silver staining was performed to identify Wx proteins using the Thermo unstained protein marker standard.

To extract total grain proteins, the developing endosperm was ground in a pestle and mortar. To extract the proteins, a previously published protocol was followed [53]. The soluble proteins were isolated in a discontinuous Tris-HCl-SDS buffer system (pH: 6.8/8.8) with a polyacrylamide concentration of 12% (*w*/*v*). After the tracer dye just migrated from the gel, electrophoresis was performed for 2 h at a constant current of 100 v/gel. The soluble proteins were detected by performing silver staining of the protein gel.

### 4.3. Cloning of the Wx-B1 Gene

The genomic DNA of both mutant and WT lines was extracted from the leaves of 2-week-old seedlings using the cetyltrimethylammonium bromide (CTAB) method [54]. As the complete genome sequence of the *Wx-1* gene is approximately 2.8 kb and showed high similarity in the A and B genomes, a segmented amplification strategy was used to obtain the complete ORF. PCR amplification was performed using Phanta Max Super-Fidelity DNA polymerase and 2 Phanta buffer (Vazyme, Nanjing, China) to avoid introducing errors into the sequence. The reaction was conducted in a total volume of 50 µL, and the PCR temperature and time were the same as those used in the previously published method. The PCR products were separated on a 1% agarose gel. After recovering the PCR products, the target DNA fragments were purified and ligated to the pBM23 Topsmart Cloning Kit (Biomed, Beijing, China). DNA sequencing was performed by Sangon Biotech (Beijing, China). The final nucleotide sequence of each fragment was determined from the sequencing results of at least three independent clones. The complete sequences of the *Wx-B1* gene were assembled and aligned using DNAman (version 9.0; Lynnon Biosoft, San Ramon, CA, USA).

### 4.4. Site-Directed Mutagenesis

To perform site-directed mutagenesis, the protein sequence of both mutant and WT lines was analyzed using the protparam tool (https://web.expasy.org/protparam, accessed on 18 January 2021). A specific position with different substitutions (G2084A) was identified in the WT *Wx-B1* gene sequence using the overlap PCR method. A single mutation vector (A469T) containing G2084A (Appendix A) was constructed. Three primer pairs (Appendix A) were designed in Primer Premier 5.0, two of which were chimeric (primers OP-F1 and OP-R1 and OP-F2 and OP-R2) for the primary PCR step. The other pair of primers (OP-F and OP-R) was nonchimeric and was used for the secondary PCR reaction.

### 4.5. Expression and Purification of GBSSI Protein

The full length Wx-B1 CDS fragment of GBSSI protein of the WT, M3-415, and A469T lines was inserted between the SacI and HindIII sites of pET-32a, located downstream of the T7 promoter, and was incorporated in the frame with the C-terminal His6 tag. Next, the expression vectors of the WT and mutant lines were transformed into *E. coli* BL21 (DE3) cells, which were grown in 50 mL of LB medium containing 50 µg ml−1 kanamycin to OD600, induced with 50 µM isopropy1 β-D-1-thiogalactopyranoside, and incubated overnight at 16 °C. The cultured bacterial cells were lysed with Bugbuster (Novagen, San Diego, CA, USA), and fusion proteins were purified on a Ni-NTA column according to the manufacturer’s instructions under native conditions (Qiagen, Dusseldorf, Germany).

### 4.6. GBSSI Starch-Binding Assay

The adsorption of recombinant GBSSI proteins to raw starch was measured through the system that is normally used to determine the binding of dextran to SBD proteins with minor modifications [55]. In addition, the resistance of recombinant GBSSI proteins to SDS was tested using previously published methods with minor modifications [56,57]. Purified recombinant GBSSI (final concentration, 5 μM) was added to a 10-μL 10% (*w*/*v*) corn starch suspension in 100 mM citrate-phosphate buffer (pH 5.0) to obtain a final volume of 60 μL. Each mixture was incubated with gentle shaking (10 rpm) at 4 °C for 1 h and was centrifuged at 12,000× *g* for 5 min at 4 °C. The upper material was designated as “supernatant′”, and in this case, it was referred to as “detergent”. The samples that were treated with SDS buffer were used to release the bound GBSSI from the starch granules, which were designated as the “bound” ones. The amount of GBSSI in the supernatant and wash and bound complexes for each assay were analyzed through Western blotting using an anti-GBSSI polyclonal antibody.

### 4.7. Real-Time Quantitative PCR (qRT-PCR) Analysis

The expression of the *Wx-B1* gene was compared between the mutant and WT lines using qRT-PCR. The analysis was conducted using primers designed for the target cDNA specific to *Wx-B1* (Appendix A), which were amplified with Chamq^TM^ Universal SYBR^®®^ qPCR Master Mix (Vazyme) on a CFX 96 Real-Time System (Bio-Rad, Hercules, CA, USA). The qRT-PCR data were analyzed in CFX Manager (Bio-Rad), and the relative expression was calculated using the 2^−^^△△Ct^ method. The relative expression of the candidate genes was normalized using the actin and GAPDH genes as reference genes.

### 4.8. GBSSI Enzymatic Activity Assay

To analyze the enzymatic activity of GBSSI, specific assays were performed based on a previously published method [58]. M3-415 and WT grains were collected 10 days after flowering. The GBSSI Enzyme Activity Assay Kit (No. BC3295) was used following the manufacturer’s instructions (Solarbio Science and Technology, Beijing, China). The assays were performed three times to cover assay error.

### 4.9. Starch Content Test

The total starch and amylose contents of the mutant and WT lines were determined using the total Amylose Assay kit and Direct Amylose/Amylose Assay kit (Megazyme, Wicklow, Ireland) according to the manufacturer’s protocol. The results of starch content in the mutant and WT lines were obtained from three repeats. Statistical analysis was performed using an independent sample Student’s *t*-test [59].

### 4.10. Determination of the Gelatinization Characteristics of Starch in Urea

For each sample, 20 mg of starch powder was mixed with 1 mL of urea solution with different molar concentrations (0, 3, 3.5, 4, 4.5, 5, 5.5, and 6 mol/L), and the mixture was incubated at a room temperature of 26 °C for 24 h. Then, the suspension was centrifuged (8000 rpm) at 26 °C for 20 min, and entire suspension was left to rest for 1 h [60,61]. To determine the degree of gelatinization, potassium iodide was added to 200 μL of supernatant and the color change was observed [62].

### 4.11. Determination of the Thermal Properties of Starch

The thermal properties of the starch paste were analyzed using a DSC (DSC 2920, TA Instruments, New Castle, DE, USA) equipped with a refrigerated cooling system. The starch samples (10 mg) were accurately weighed using aluminum Tzero discs (TA Instruments) and mixed with 20 μL of deionized water in a 1:2 starch:water ratio. The discs were sealed and equilibrated at room temperature for 2 h. Within a temperature range of 30–100 °C, the heating rate was 10 °C min^−1^. Empty indium discs were used as reference standard calibration instruments. Enthalpy of gelatinization (ΔH), onset temperature (To), peak temperature (TP), and completion temperature (TC) were measured and calculated using the Universal Analysis 2000 V 4.7 A software (TA Instruments).

### 4.12. Analysis of Starch Granule Morphology

The granular morphology of starch was examined using an SEM [63]. The extracted starch and distilled water were mixed in a ratio of 1:100. Then, 20 μL of the starch solution was transferred to an aluminum Tzero plate (TA Instruments, ASSE, Asse, Belgium) and was left to dry naturally at a room temperature of 26 °C. The samples were photographed with Fei Quanta 450 (Fei Corporation, Hillsboro, OR, USA) and were subsequently examined. For each sample, three different layer positions were selected to assess the morphology of the starch granules at different magnification levels. The starch granules were divided into A- (>10 μm) and B-type (1–10 μm) based on their diameter, and the number and volume percentages of each type in mutants were calculated using ImageJ (NIH, Bethesda, MD, USA) (bundled with a 64-bit Java v1.8.0) as previously described with some modifications [64]. The granule size of the starch slurries was measured using Mastersizer 3000 (Malvern Instruments, Malvern, UK) according to the manufacturer’s instructions.

## 5. Conclusions

This study identified a new mechanism affecting GBSSI activity and amylose production in tetraploid wheat using EMS. The mutation identified in the present study is different from those previously reported, and we found that a single amino acid change in the ADPG binding pocket can regulate starch-binding capabilities. Based on this finding, we herein clarified the correlation among the structure of GBSSI protein, variation of its activity, and alteration in amylose content. We concluded that achieving low amylose contents is possible by editing the functional domains of GBSSI protein. In addition, the results of our study provide a theoretical basis for breeding wheat cultivars with different amylose contents to meet diverse consumer demands.

## Figures and Tables

**Figure 1 ijms-23-08432-f001:**
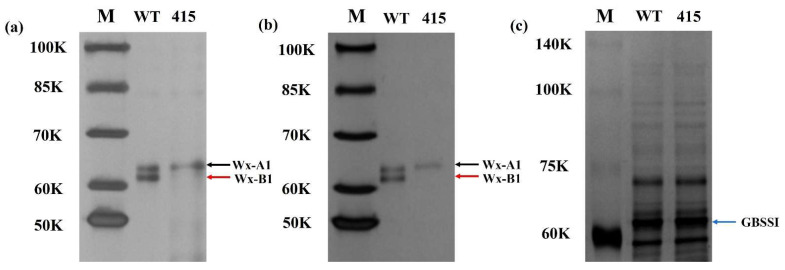
Profiles of the Wx proteins and total grain proteins obtained via SDS-PAGE. Wx-A1 and Wx-B1 proteins in mature (**a**) and developing (**b**) grains are represented by black and red arrows. The total grain proteins (**c**) were extracted from the WT and mutant 415. lines, the blue arrow represents GBSSI protein in both lines.

**Figure 2 ijms-23-08432-f002:**
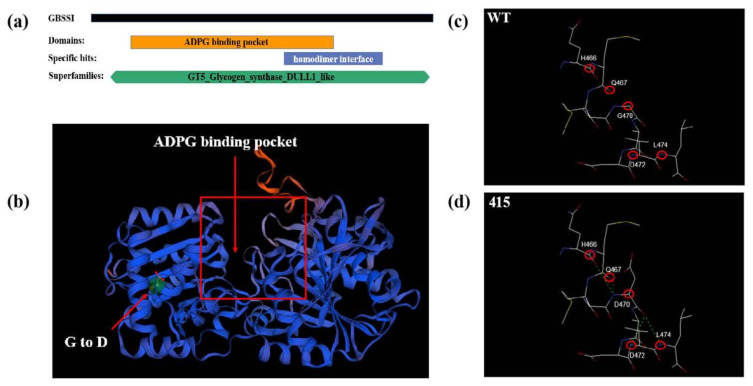
Protein structure of GBSSI representing the position of specific domain and representative impacts of amino acid residue substitutions at 470 on residue accessibility to solvent and hydrogen bond forces surrounding the amino acid residues. (**a**) NCBI analysis predicting the function domains in tetraploid wheat GBSSI sequences. (**b**) The Swiss-pdb viewer required for ADPG binding in GBSSI were homology modeling, and it was derived using an EcGS structure (PDB, 3GUH) as a model. The ADPG binding pocket is shown by the red box. The mutation point is shown by the red arrow near the ADPG binding pocket representing the G-to-D mutation. (**c**) Protein structure with the original amino acid G470. (**d**) Substitution of amino acid D470 and its effects on the protein structure. Residues are shown in red, turquoise, or blue, which indicate their solvent accessibility ranging from high to low. Hydrogen bonds between the residues of interest and their surrounding atoms are shown. The green dashed line indicates strong hydrogen bonds. In addition, the length of the hydrogen bonds connecting the atoms is also shown.

**Figure 3 ijms-23-08432-f003:**
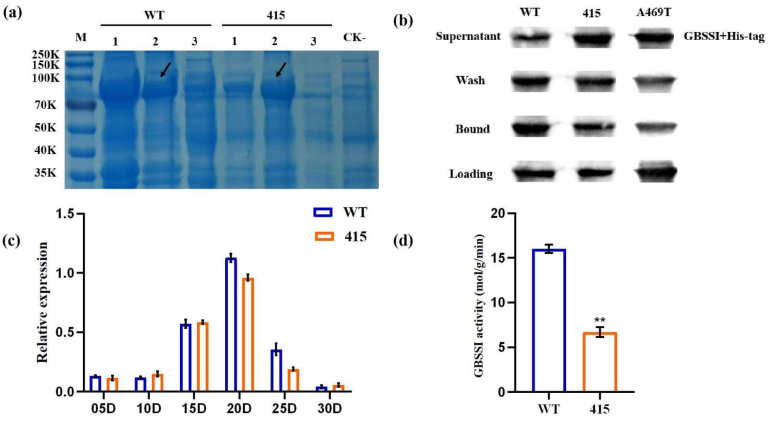
Expression of GBSSI protein and binding ability of mutated recombinant GBSSI to raw starch and analysis of the GBSSI enzyme activity and expression levels of wheat *Wx-B1*.(**a**) GBSSI activity was assayed 10 d after fertilization (DAF) of the endosperms. Asterisks indicate statistical significance between the WT and mutant lines, as determined using the Student’s *t*-test (** *p* < 0.01). (**b**) The expression levels for each accession in the parental WT and mutant M3-415 lines at 5–30 days after anthesis (DAA) in tetraploid wheat developing grains were calibrated as expression folds compared to those at 3 DAA. The *x*-axis and *y*-axis represent relative expression and DAA, respectively. (**c**) Expression of GBSSI protein in the supernatant to obtain the purified recombinant protein. Lanes are described as follows: (1) whole bacteria; (2) supernatant; and (3) precipitate. Black solid arrows indicate GBSSI protein in the supernatant. (**d**) Ability of mutated recombinant GBSSI to bind to raw starch. Corn starch was incubated with purified recombinant GBSSIs. The original unbound GBSSI was designated as “supernatant”. After SDS washing and centrifugation, the supernatant was defined as “wash”. Insoluble GBSSIs were defined as “bound”. The amounts of each GBSSI loaded on the gel per treatment were equivalent.

**Figure 4 ijms-23-08432-f004:**
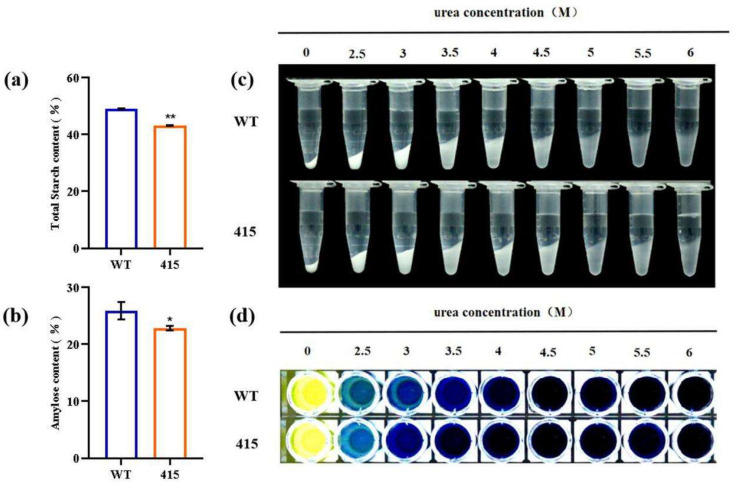
Starch parameters and characteristics of starch gelatinization in urea solution for the parental WT and mutant M3-415 lines. (**a**,**b**) Total starch and amylose contents of the WT and M3-415 lines. Values are expressed as means ± SD (*n* = 3). Asterisks indicate statistical significance between the lines as determined using the Student’s *t*-test (* *p* < 0.05; ** *p* < 0.01). (**c**,**d**) Starch in WT and M3-415 grains was difficult to gelatinize in urea solution. The starch powder was mixed with different concentrations (0–6 M) of urea solution, and distilled water was used a blank control.

**Figure 5 ijms-23-08432-f005:**
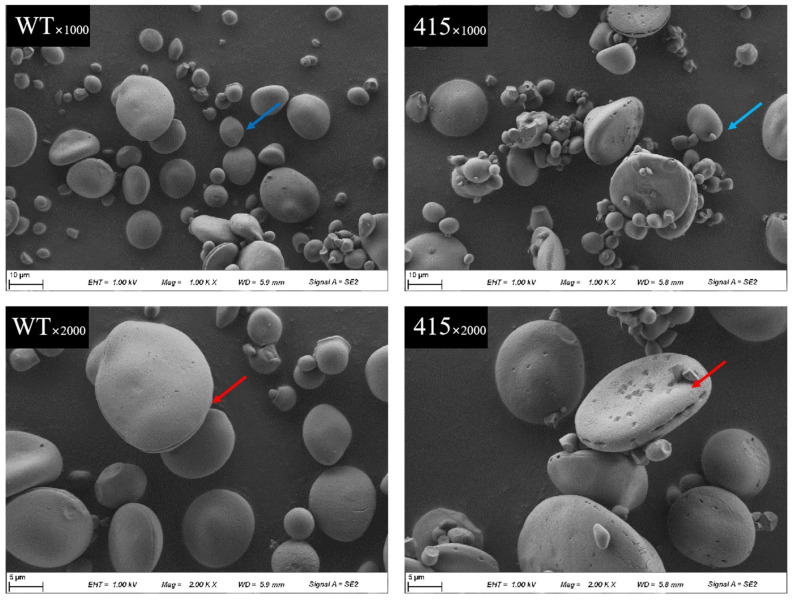
Starch granule structure in the parental WT and mutant M3-415 lines analyzed via SEM. The red and blue arrows indicate the A-type and B-type starch granules, respectively.

**Table 1 ijms-23-08432-t001:** The prediction results of Wx-B1 protein.

Protein	SAAS	Predicted Score ^a^	Predicted Class	Observed Class
Wx-B1	G470D	0.999	Functional	Functional

SAAS, single amino acid substitutions. ^a^ When predicted score ≥ 0.5, the predicted class and the observed class are predicted to be functional, and when predicted score < 0.5, the predicted class and the observed are predicted to be neutral.

**Table 2 ijms-23-08432-t002:** Analysis of the thermal properties of starch in the WT and M3-415 lines.

Accession	To (°C)	Tp (°C)	Tc (°C)	ΔH (J/g)
WT	62.73 ± 0.56	65.54 ± 0.48	76.73 ± 0.59	3.607 ± 0.28
415	62.49 ± 0.12	67.22 ± 0.28 **	77.93 ± 0.36 **	6.262 ± 0.39 **

DSC measured 1:2 (*w*/*w* db) starch:water ratio. Triplicate measurements were taken for each sample and expressed as mean ± SD. The asterisks indicate the statistical significance (** *p*  <  0.01).

**Table 3 ijms-23-08432-t003:** Statistical analysis of starch granule size.

	Granule Content—Number (%)		Granule Content—Volume (%)	
Line	A-type	B-type	A-type	B-type
WT	0.23	99.74	56.84	35.29
415	0.23	99.79	58.58 *	39.13 **

The starch granules were divided into large A- (diameter > 10 μm) and small B-type (diameter < 10 μm). The granule content of starch were calculated by granule numbers (in the second column) and granule volumes (in the third column), respectively. Independent sample Student’s *t*-test was used for data analysis, the statistical significance was indicated as * *p*  <  0.05; ** *p*  <  0.01.

## Data Availability

The study did not report any data.

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
