# Peer review of "Regulation of Amylose Content by Single Mutations at an Active Site in the Wx-B1 Gene in a Tetraploid Wheat Mutant"

_ijms, 2022, doi:10.3390/ijms23158432_

Round 1

Reviewer 1 Report

This could be a nice paper that shows a different way of getting an effective "null" mutant in a Triticum waxy gene.  I have suggested major revision   of English expression and some other things. I think the native English writer can do this, with the passage of time from the last version submitted, and re-looking at this. 

Some examples (only) of English are:

"Wheat is one of the world's top ten essential crops [1]." would be better as "Wheat is a crop of major world significance [1]."

“Starch is the most dominant component of wheat grains accounting for approximately 70% of the total dry grain weight [5]” – as it is 70%, is must be the dominant component. Just say “Starch is the major component of wheat grain at 70% of the dry weight”

Paragraph 2 of the Introduction. “The starch biosynthesis pathway is a complex mechanism that involves multiple enzymes and isoforms.” Would be better as “Starch synthesis is complex involving multiple enzymes and isoforms.” (As a scientific aside, I don’t think there is any synthesis in plants that is not complex involving multiple enzymes and isoforms, so starch is not “special”.) Use “has” for “exhibits”.

Paragraph 3 of the Introduction. “Researchers showed …” should be something like “The sequence of the Wx-B1 gene of the mutant …”. (Later in the manuscript, the word “researchers” is used a number of times in different contexts; these should be removed and replaced with correct word usages.)

Some other things:

Figs 2 and 3 are in the wrong order.

Fig 1 legend,  the red arrow also indicates GBSS protein in WT

Figs 3 and 4, the blue and orange fill in the histograms should be removed, these colours add no information to the figures, and do, in fact, obscure the error bars

Table 2, the results in this table are also given in full in the text, lines 190-192 - do one or the other, and the foot note to the table has the meanings of *. and **, but these are not in the table. 

Table 1, there is an "a" but no explanation of what this indicates; the work predicted is used 2x more that needed in the footnote

4.8, line 397, need to make clearer as "were performed three times to cover assay error." This is a technical replication, not replication of taking the samples from the plants, or of replicated plants of the genotypes.

In various places T-tests are referred to, as "t-test", "T test", "Student's t-test", 

Author Response

  1. English needs to be improved.

Response: We have employed a commercial company edit service to improve the language.The detailed modifications were highlighted in track change.

  1. "Wheat is one of the world's top ten essential crops [1]." would be better as "Wheat is a crop of major world significance [1]."

Response: We have revised the sentence as the reviewer suggested.

Line 35: Wheat is a crop of major world significance[1]. In particular, tetraploid durum wheat (Triticum turgidum L. ssp. durum (Desf.) Husn, 2n = 4x = 28, AABB) is an important cereal crop with an annual worldwide production of over 40 million tons [2].

  1. “Starch is the most dominant component of wheat grains accounting for approximately 70% of the total dry grain weight [5]” – as it is 70%, is must be the dominant component. Just say “Starch is the major component of wheat grain at 70% of the dry weight”.

Response: We have modified the sentence as the reviewer suggested.

Line 39: Starch is the major component of wheat grain at 70% of the dry weight [5], consists of two classes of glucose polymers, amylose and amylopectin, which are defined based on their α-1, 4 and α-1, 6 linkages and molecular weight. Amylose features a low degree of polymerization (DP < 5000) as compared with amylopectin (DP > 5000) [6, 7]. The amylose to amylopectin ratio determines some of the functional and physicochemical of starch, thereby affecting the quality of durum wheat [8, 9].

  1. Paragraph 2 of the Introduction. “The starch biosynthesis pathway is a complex mechanism that involves multiple enzymes and isoforms.” Would be better as “Starch synthesis is complex involving multiple enzymes and isoforms.” (As a scientific aside, I don’t think there is any synthesis in plants that is not complex involving multiple enzymes and isoforms, so starch is not “special”.) Use “has” for “exhibits”.

Response: Done.

Line 44: Starch synthesis is a complex process involving multiple enzymes and isoforms. The synthesis of amylopectin requires three important classes of enzymes: starch synthase (SS), starch branching enzymes (SBE), and starch debranching enzymes (DBE)[6]. Amylose synthesis requires only granule-bound starch synthase I (GBSSI or waxy protein) enzyme [10, 11]. Durum wheat has two waxy proteins encoded by Wx-A1 and Wx-B1 [12].

  1. Paragraph 3 of the Introduction. “Researchers showed …” should be something like “The sequence of the Wx-B1 gene of the mutant …”. (Later in the manuscript, the word “researchers” is used a number of times in different contexts; these should be removed and replaced with correct word usages.).

Response: We have changed this sentence according to the reviewer suggestion.

Line 64: The sequence of the Wx-B1 gene of the mutant can encoded a complete protein, which was incompatible with the protein profile obtained via sodium dodecyl-sulfate polyacrylamide gel electrophoresis (SDS-PAGE) showing the lack of Wx-B1 protein in the mutant line.

Line 99: Thus, we assumed that Wx-B1 might be present in the grains, as evidenced by the results of total grain protein, but without being bound to starch granules.

Line 105: We hypothesized that the mutated Wx-B1 protein may have lost the ability to bind to starch granules.

Line 114: Therefore, we speculated that the likely effect of the mutation at this site was to prevent GBSSI from binding to starch granules

Line 257: Here we proposed that the M3-415 mutant, as a partial waxy durum wheat, might contribute to a better end-use quality of pasta.

Some other things:

  1. Figs 2 and 3 are in the wrong order.

Response: We have adjusted the order of the Figure.

  1. Fig 1 legend, the red arrow also indicates GBSS protein in WT.

Response: We have replaced the red arrow with the blue arrow and modify the figure legend to clearly indicate the GBSS protein in Fig 1c according to the reviewer suggestion.

  1. Figs 3 and 4, the blue and orange fill in the histograms should be removed, these colors add no information to the figures, and do, in fact, obscure the error bars.

Response: We have remade the Figs 3 and 4 to clearly show the error bars.

  1. Table 2, the results in this table are also given in full in the text, lines 190-192 - do one or the other, and the foot note to the table has the meanings of *. and **, but these are not in the table. Table 1, there is an "a" but no explanation of what this indicates; the work predicted is used 2x more that needed in the footnote.

Response: To avoid repeat, we have removed the detailed descriptions of starch thermal properties in line 219 and just keep the Table 2. We also corrected the mistake of miss * and **, the symbol were indicated to show in the table. In addition, we have re-tested the starch thermal properties to give accurate data according to the reviewer’s suggestion. The accurate data were summarized in Table 2. The corresponding descriptions were listed in the sections of “Results” and “Discussion” at lines 164-166 and 240-245:

Table 2. Analysis of the thermal properties of starch in the WT and M3-415 lines.

Accession

To(°C)

Tp(°C)

Tc(°C)

ΔH(J/g)

WT

62.73±0.56

65.54±0.48

76.73±0.59

3.607±0.28

415

62.49±0.12

   67.22±0.28** 

   77.93±0.36 **

   6.262±0.39**

DSC measured 1:2 (w/w db) starch:water ratio. Triplicate measurements were taken for each sample and expressed as mean ± SD. The asterisks indicate the statistical significance (**p < 0.01).

We have corrected the colum of “a” at “Predicted score”, and explained the "a" in Table 1 to avoid misleading reader according to th reviewer suggestion.

Table 1. The prediction results of Wx-B1 protein.

Protein

SAAS

Predicted scorea

Predicted class

Observed class

Wx-B1

G470D

0.999

Functional

Functional

SAAS, single amino acid substitutions. aWhen predicted score ≥ 0.5, the predicted class and Observed class are predicted to be functional, and when predicted score < 0.5, Predicted class and Observed class are predicted to be neutral.

  1. 4.8, line 397, need to make clearer as "were performed three times to cover assay error." This is a technical replication, not replication of taking the samples from the plants, or of replicated plants of the genotypes.

Response: We have revised the sentence as the reviewer suggested.

Line 331: To analyze the enzymatic activity of GBSSI, specific assays were performed based on a previously published method [58]. M3-415 and WT grains were collected 10 days after flowering. The GBSSI Enzyme Activity Assay Kit (No. BC3295) was used following the manufacturer’s instructions (Solarbio Science & Technology, Beijing, China). The assays were performed three times to cover assay error.

  1. In various places T-tests are referred to, as "t-test", "Ttest", "Student's t-test".

Response: Done.

Line 152: Asterisks indicate statistical significance between the WT and mutant lines, as determined using the Student’s t-test (*P < 0.05; **P < 0.01). (d) The expression levels for each accession in the parental WT and mutant M3-415 lines at 5–30 days after anthesis (DAA) in tetraploid wheat developing grains were calibrated as expression folds compared to those at 3 DAA. The x-axis and y-axis represent relative expression and DAA, respectively.

Line 171:Values are expressed as means ± SD (n = 3). Asterisks indicate statistical significance between the lines as determined using the Student’s t-test (*P < 0.05; **P < 0.01).

Line 193: Independent sample Student’s t-test was used for data analysis, the statistical significance was indicated as * p < 0.05; ** p < 0.01.  

Line 336: Statistical analysis was performed using an independent sample Student’s t-test [59]. 

Reviewer 2 Report

Manuscript is well written, and I suggest a few minor editing points

Line 58: GBSSI, soluble starch synthase, and glycogen synthase (GS) all belong to the GT5 (GT-B) glucosyltransferase family and share a common structure consisting in the C-terminal and N-terminal domains containing a donor and acceptor binding site, respectively. "-- Consisting seems out of place perhaps existing?

Line 79: “An M2 mutagenized population derived from cultivar LM47 was screened  -- M2 not identified at first occurrence.

Line 97: “led to a residue change from glycine to the aspartic in the deduced protein.. .” change the aspartic  to aspartic acid.

Line 145: “an enhanced van der Waals force was produced at D470 to baffle the ADP …” Is baffle correct term perhaps block or hinder.   

Line 146: “Therefore, researchers speculated that the mutation

Line 149: “position of special domain…”  Special or specific domain?

Line 214: “whereas the volumes of the A- and B- type starch granules 214 in the M3-415 line were larger than those in the WT line (Table 3).” But not statistically significant difference – No P value is indicated. Also this difference is cited in Line 284

“In this study, the thermodynamic temperature of the M3-415 mutant was lower in the mutant line than in the WT line, which is not in line with the findings reported in previous studies.”

Line 230: compact wheat should be identified as T. compactum or club wheat.

Line 286: I would probably refer to the rough microstructure as pitting. 

Figure 3 description and placement precedes Fig 2. Fig 2 and 3 should be switched so that they appear in order in the text and manuscript.

Table 3 The third column (Granule content) has no units

Author Response

  1. Line 58: GBSSI, soluble starch synthase, and glycogen synthase (GS) all belong to the GT5 (GT-B) glucosyltransferase family and share a common structure consisting in the C-terminal and N-terminal domains containing a donor and acceptor binding site, respectively. "-- Consisting seems out of place perhaps existing?

Response: According to your suggestion, we have replaced consisting with existing.

Line 58: GBSSI, soluble starch synthase, and glycogen synthase (GS) belong to the GT5 (GT-B) glucosyltransferase family. These proteins share a common structure existing in the C-terminal and N-terminal domains containing a donor- and acceptor- binding site, respectively [17-19].

  1. Line 79: “An M2 mutagenized population derived from cultivar LM47 was screened -- M2 not identified at first occurrence.

Response: We have revised this mistake through the whole manuscript according to the reviewer’s suggestion.

Line 79 : The M2-mutagenized population of cultivar LM47 was screened for variability in waxy proteins using SDS-PAGE. Three grains were chosen from each plant from the mutagenized population, and M2-415 was identified as a mutant lacking Wx-B1 protein.

  1. Line 97: “led to a residue change from glycine to the aspartic in the deduced protein.. .” change the aspartic to aspartic acid.

Response: We have changed the aspartic to aspartic acid from the manuscript according to the reviewer's suggestion.

Line 97 : This mutation changed the codon from GGC to GAC and led to a residue change from glycine to aspartic acid in the deduced protein sequence; however, a stop codon was not observed (Figure S1).

Response: We have changed the compact wheat to aspartic acid from the manuscript according to the reviewer's suggestion.

  1. Line 145: “an enhanced van der Waals force was produced at D470 to baffle the ADP …” Is baffle correct term perhaps block or hinder.

Response: Done

Line145 : Most significantly, an enhanced van der Waals force was produced at D470, hindering the entry of ADP into the binding pocket.

  1. Line 146: “Therefore, researchers speculated that the mutation.

Response: We have revised the sentence as the reviewer suggested.

Line 146 : Therefore, We speculated that the likely effect of the mutation at this site was to prevent GBSSI from binding to starch granules.

  1. Line 149: “position of special domain…” Special or specific domain?

Response: We have changed this word as the reviewer required

Line 149 : Figure 2. Protein structure of GBSSI representing the position of specific domain and representative impacts of amino acid residue substitutions at 470 on residue accessibility to solvent, and hydrogen bond forces surrounding the amino acid residues.

  1. Line 214: “whereas the volumes of the A- and B- type starch granules 214 in the M3-415 line were larger than those in the WT line (Table 3).” But not statistically significant difference – No P value is indicated. Also this difference is cited in Line 284.9.

Response: We have added the P value in Table 3

Table 3. Statistical analysis of starch granule size.

Granule content – Number (%)

Granule content – Volume (%)

Line

A-type

B-type

A-type

B-type

WT

0.23

99.74

56.84

35.29

415

0.23

99.79

 58.58*

  39.13**

The starch granules were divided into large A- (diameter > 10 μm) and small B- type (diameter < 10 μm). The granule content of starch were calculated by granule numbers(in the second column) and granule volumes(in the third column), respectively. Independent sample Student’s t-test was used for data analysis, the statistical significance was indicated as * p < 0.05; ** p < 0.01.

  1. “In this study, the thermodynamic temperature of the M3-415 mutant was lower in the mutant line than in the WT line, which is not in line with the findings reported in previous studies.”

Response: We have re-conducted the DSC experiments to verify the results of starch physicochemical properties. The accurate data were summarized in Table 2. The corresponding descriptions were listed in the sections of “Results” and “Discussion” at lines 164-167 and 240-245:

Previous studies revealed that enthalpy of gelatinization (ΔH) is positively correlated with the the volumes of starch granules [45]. In this study, the M3-415 has lower amylose content, higher peak temperature and gelatinization enthalpy, this can be explained that the volumes of A-type starch granules and B-type starch granules in the mutatn line were significantly increased than those in the WT line. In addition, the presence of pitting in the microstructure of the M3-415 starch granules, this has also be observed in waxy barley [46].

 Table 2. Analysis of the thermal properties of starch in the WT and M3-415 lines.

Accession

To(°C)

Tp(°C)

Tc(°C)

ΔH(J/g)

WT

62.73±0.56

65.54±0.48

76.73±0.59

3.607±0.28

415

62.49±0.12

   67.22±0.28** 

   77.93±0.36 **

   6.262±0.39**

DSC measured 1:2 (w/w db) starch:water ratio. Triplicate measurements were taken for each sample and expressed as mean ± SD. The asterisks indicate the statistical significance (**p < 0.01).

  1. Line 230: compact wheat should be identified as T. compactumor club wheat.

Line 230: Other Wx-B1 alleles, such as Wx-B1k and Wx-B1m, were found in T. compactum and Indian dwarf wheat.

  1. Line 286: I would probably refer to the rough microstructure as pitting.

Response: Done.

Line 286: In addition, the presence of pitting in the microstructure of the M3-415 starch granules, this has also be observed in waxy barley [46].

  1. Figure 3 description and placement precedes Fig 2. Fig 2 and 3 should be switched so that they appear in order in the text and manuscript.

Response: We have adjusted the order of the Figure.

  1. Table 3 The third column (Granule content) has no units.

Response: We have added the column units in Table 3. The granule content of starch were calculated by granule numbers(in the second column) and granule volumes(in the third column), thus the units for second and third columns were number and volume of starch granules, respectively.

Granule content – Number (%)

Granule content – Volume (%)

Line

A-type

B-type

A-type

B-type

WT

0.23

99.74

56.84

35.29

415

0.23

99.79

  58.58**

  39.13**

The starch granules were divided into large A- (diameter > 10 μm) and small B- type (diameter < 10 μm). The granule content of starch were calculated by granule numbers(in the second column) and granule volumes(in the third column), respectively. Independent sample Student’s t-test was used for data analysis, the statistical significance was indicated as * p < 0.05; ** p < 0.01.

Round 2

Reviewer 1 Report

The authors and the English editors have done a very good job of making the changes; ready for press.